# Recombinant Globular Domain of TcpA Pilin from *Vibrio cholerae* El Tor: Recovery from Inclusion Bodies and Structural Characterization

**DOI:** 10.3390/life12111802

**Published:** 2022-11-07

**Authors:** Victor Marchenkov, Elena Dubovitskya, Nina Kotova, Igor Tuchkov, Nina Smirnova, Natalia Marchenko, Alexey Surin, Vladimir Filimonov, Gennady Semisotnov

**Affiliations:** 1Institute of Protein Research, Russian Academy of Sciences, Institutskaya Street 4, 142290 Pushchino, Russia; 2Russian Antiplague Scientific Research Institute “Microbe” of Rospotrebnadzor, Universitetskaya Street 46, 410005 Saratov, Russia; 3Branch of the Shemyakin–Ovchinnikov Institute of Bioorganic Chemistry, Russian Academy of Sciences, Prospekt Nauki 6, 142290 Pushchino, Russia; 4State Research Centre for Applied Microbiology and Biotechnology, 142279 Obolensk, Russia

**Keywords:** inclusion bodies, protein purification, protein refolding, protein thermodynamics, *Vibrio cholerae* pilin

## Abstract

The production of recombinant proteins in *Escherichia coli* cells is often hampered by aggregation of newly synthesized proteins and formation of inclusion bodies. Here we propose the use of transverse urea gradient electrophoresis (TUGE) in testing the capability of folding of a recombinant protein from inclusion bodies dissolved in urea. A plasmid encoding the amino acid sequence 55–224 of TcpA pilin (C-terminal globular domain: TcpA-C) from *Vibrio cholerae* El Tor enlarged by a His-tag on its N-terminus was expressed in *E. coli* cells. The major fraction (about 90%) of the target polypeptide was detected in cell debris. The polypeptide was isolated from the soluble fraction and recovered from inclusion bodies after their urea treatment. Some structural properties of the polypeptide from each sample proved identical. The refolding protocol was developed on the basis of TUGE data and successfully used for the protein large-scale recovery from inclusion bodies. Spectral, hydrodynamic, and thermodynamic characteristics of the recombinant TcpA recovered from inclusion bodies indicate the presence of a globular conformation with a pronounced secondary structure and a rigid tertiary structure, which is promising for the design of immunodiagnostics preparations aimed to assess the pilin level in different strains of *V. cholerae* and to develop cholera vaccines.

## 1. Introduction

Cholera, an extremely serious infectious disease, continues to pose a constant threat to the population of developing countries around the world. All pandemics have been caused by toxigenic strains of the *Vibrio cholerae* (*V. cholerae*) O1 serogroup, but different biovars, classical and El Tor, which differ in their phenotypic and molecular-genetic properties [1,2]. *V. cholerae* of the classical biovar was probably the causative agent of the first six pandemics from 1817 to 1923; later, it was superseded by *V. cholerae* biovar El Tor causing the current seventh cholera pandemic that started in 1961 [3].

At present, it has been established that the *V. cholera* El Tor agent gene *tcpA* is represented by two main alleles: *tcpA^ET^* and *tcpA^CIRS^* [4,5]. The *tcpA^ET^* allele is typical for pathogenic strains that caused cholera epidemics in 1961 to 2002, while *tcpA^CIRS^* allele was found later in the highly pathogenic strains causing the 2003–present day epidemics in many regions, including Russia [4,5]. Unlike *tcpA^ET^*, *tcpA^CIRS^* carries one nucleotide substitution of adenine for guanine (A/G) at position 266 of the *tcpA* gene, which leads to a change in the TcpA aminoacid sequence at position 89 where asparagine is replaced by serine (Appendix A, alignment by BLAST [6]). The functional significance of this substitution has not been precisely determined yet. Nevertheless, the wide distribution of the pathogenic gene variants with such a modification in the *tcpA* structure within different regions of the world (including the territory of Russia in recent years) is likely due to the displacement of other strains of the cholera pathogen. This was the main reason for our choice of the *tcpA^CIRS^* allele for creating a strain-producer of the encoded protein. This work is aimed to construct a recombinant strain of *E. coli* which is the producer of the TcpA protein of the cholera El Tor agent carrying the *tcpA^CIRS^* gene.

To date, several recombinant strains of *E. coli* are known to be designed to produce the TcpA protein encoded by different alleles (but not *tcpA^CIRS^*) for the subsequent creation of cholera vaccines on their basis [7,8,9,10]. However, in the Russian Federation, there is no such strain-producer of TcpA needed to have both domestic recombinant cholera vaccines of new generation and the immunodiagnostic test system required to assess the level of pathogenicity produced by epidemically dangerous strains.

The pathogenicity of a cholera agent, regardless of the biovar, is provided by the action of two biomolecules that are the main virulence factors, namely the toxin-coregulated pilus (TCP) and cholera toxin (CT). The major subunit of TCP is coded by the nucleotide sequence of the *tcpA* gene (672 nucleotide pairs), which in El Tor is only 82% identical to the classical vibrios [11].

This study is focused on TcpA pilin which plays an important role in the pathogenesis of cholera [12,13]. Toxigenic strains of *V. cholerae* that do not produce this protein are unable to colonize the human small intestine and to develop the subsequent infection [12]. This underlies the design of TcpA-based immunodiagnostic preparations aimed to assess the level of TCP expression in modern variants of the pathogen and to determine their ability to initiate the cholera infection by colonizing the small intestine. Moreover, it has been proven that TcpA is one of the key protective antigens whose antibodies are able to block the attachment of *V. cholerae* to epithelial cells of the small intestine, thus preventing the development of cholera infection [14,15]. All these facts indicate that the protein TcpA can be used to develop cholera vaccines.

It is known that the entire TcpA polypeptide synthesized in *V. cholerae* contains 224 amino acids [11,13]. The first 25 amino acid residues belong to the signal peptide that is cleaved off upon protein transfer into the periplasm. The remaining 199 amino acid residues correspond to the “mature” pilin (mTcpA) forming pili. The first 52 residues of mTcpA fold into a long α-helix (α1) present in all pilins of the IVb class. This long helix can be subdivided into two segments, α1-N and α1-C [13,16]. The N-terminal segment (α1-N, 28 amino acids long) responsible for pilin oligomerization has a highly conserved and hydrophobic primary structure. In contrast, α1-C belongs to the globular C-terminal domain of TcpA (about 171 amino acids in total) with a high content of the secondary structure and the sole disulfide bond between Cys145 and Cys211. α1-N participates in polymerization and interferes with protein crystallization; therefore, only the globular C-terminal domain of TcpA (TcpA-C) was used as a recombinant protein, purified, and crystallized. Its 3D structure was determined at a resolution of 1.3–1.6 Å [13,16].

We decided to develop our own protocol of TcpA-C (*tcpA^CIRS^* gene) large-scale purification and to characterize its physical and chemical properties in solution for the purpose of its subsequent use in antibody preparation.

## 2. Materials and Methods

### 2.1. Bacterial Strains and Cultivation Conditions

Previously, we constructed a recombinant strain *Escherichia coli* BL21 Star^TM^(DE3)/pET302tcpA (*E. coli* BL21 Star^TM^ from Thermo Fisher Scientific, Waltham, MA, USA) using the plasmid expressing a part of the *tcpA^CIRS^* gene fused with His-Tag (total length 534 bp) that was placed under the transcriptional control of the phage T7 promoter [17] (Appendix A). The donor of the *tcpA^CIRS^* gene was the strain *V. cholerae* M1429 biovar El Tor that is an atypical variant of the pathogen isolated from a Russian patient in 2005 (NCBI GenBank accession number: LAEM00000000.1, https://www.ncbi.nlm.nih.gov/nuccore/LAEM00000000.1 (accessed on 5 November 2022)). The polymerase chain reaction (PCR) and calculated primers (F ccgctcgaggattcgcagaatatgactaaggctgc and R gctggatccttaactgttaccaaaagctactgtgaat) were used to incorporate the *tcpA^CIRS^-C* fragment into the expression plasmid pET302. The nucleotide sequence of *tcpA^CIRS^-C* was checked and found to fit the known data on the *tcpA^CIRS^* gene [4]; the sequencing was performed using an ABI3500xl automated sequencer (Applied Biosystems, Waltham, MA, USA) using BigDye^®^ Terminator v3.1 Cycle Sequencing Kit. In our vector, the corresponding N-terminal polypeptide sequence was fMHHHHHH-S55-Q56-N57-M-58-… (Appendix A), so the expressed polypeptide TcpA-C was 177 amino acids long for lack of N-terminal 25 a.a. of the leader peptide and 28 a.a. of the α1-N segment responsible for the pilin polymerization (Appendix A). The molecular weight of this polypeptide computed at https://web.expasy.org/protparam/ (accessed on 5 November 2022) [18] was 18.2 kDa. The bacteria were cultured in broth and agar LB medium, pH 7.4, at 30 °C or 37 °C without antibiotics or with ampicillin (50 µg/mL). Isopropyl-β-D-thiogalactoside (IPTG) (Invitrogen, Waltham, MA, USA) was used as an inductor of expression of the cloned *tcpA^CIRS^* gene [17]. The optimal conditions for cultivating the recombinant strain were the liquid medium LB (pH 7.4), 50 µg/mL ampicillin, enhanced aeration (200 rpm), 37 °C, 2 h, and an IPTG concentration of 0.1 mM. 

### 2.2. Preparation of Cell Lysates

*E. coli* cells producing the target protein were precipitated by centrifugation at 10,000× *g* for 10 min, washed, and resuspended in a cold buffer with 10 mM Tris-HCl, pH 8.0, 0.3 M NaCl. The precipitated cells were lysed using an ultrasonic Branson 450 Digital Sonifier homogenizer (Branson Ultrasonic Corporation, Danbury, CT, USA) with ten cycles of 30 s ON and 30 s OFF at a frequency of 60 kHz in an ice bath. The separation of soluble and insoluble fractions of TcpA-C was carried out by centrifugation of the broken cells at 10,000× *g* for 10 min. The presence of protein in the precipitant and the supernatant was monitored by SDS-PAGE electrophoresis.

### 2.3. Purification of the Protein from the Soluble Fraction

The bulk of TcpA-C (about 90%) was detected in inclusion bodies. The remaining ~10% of the protein population localized to the soluble fraction of cell lysate and was purified in two steps: (1) by affinity chromatography on Ni-NTA resin (Profinia™ Affinity Chromatography Protein Purification System; Bio-Rad Laboratories, Inc., Hercules, CA, USA), followed by (2) size-exclusion chromatography on a Superdex 200 column (GE Healthcare, Chicago, IL, USA), as described elsewhere [17]. At the first stage of purification, the soluble fraction of cell lysate was applied to the Ni-NTA column. Washing was carried out using Native IMAC Wash Buffer (Bio-Rad Laboratories, Inc., Hercules, CA, USA) with stepwise increasing concentrations of imidazole—30 mM (10.0 mL), 50 mM (10.0 mL), and 100 mM (10.0 mL)—to reduce nonspecific binding of contaminating proteins to the sorbent and increase the purity of the TcpA protein. Native IMAC Elution Buffer (Bio-Rad Laboratories, Inc., Hercules, CA, USA) with 250 mM imidazole was used in elution. At the second stage, size-exclusion chromatography on a Superdex 200 column (GE Healthcare, Chicago, IL, USA) used 20 mM Tris-HCl buffer, pH 8.9, and 150 mM NaCl. The peak containing the protein of interest with *M_r_* of about 20 kDa was collected, concentrated up to 0.5 mg/mL, and used for the subsequent analysis.

### 2.4. Recovery of the Target Protein from Inclusion Bodies through the Urea-Unfolded State

The inclusion bodies were thoroughly washed two times with 20 mM Tris-HCl, pH 8.9, 1% Deoxycholate; two times with 20 mM Tris-HCl, pH 8.9, 1.0 M urea; and six times with 20 mM Tris-HCl, pH 8.9. The washing was performed by resuspension in the corresponding buffer, followed by centrifugation at 12,100× *g* for 5 min, and led to essential removal of the protein impurities containing in the cell debris. 

To remove the few remaining protein contaminants, we used Ni-NTA chromatography in 8.0 M urea, as the first step. TcpA-C fused with the His-tag has an isoelectric point of about 8.0 (which is close to the pH optimum of the resin); therefore, we raised the buffer pH up to 8.9 to increase the protein solubility during its refolding. This resulted in the increased non-specific protein binding to Ni-NTA, making it very difficult to remove some impurities from the protein preparations. After Ni-NTA chromatography, protein-containing fractions were collected and loaded on a Mono Q 5/50GL column, as the second step of the domain purification. The fractions enriched with the target protein were collected and concentrated to 150 mkl using an Amicon centrifugal concentrator MWCO 10 kDa (Millipore Ireland Ltd., Tullagreen, County Cork Carrigtwohill, Ireland). The protein renaturation was performed by 20-fold dilution on ice, with intensive stirring, using the renaturation buffer optimized with a QuickFold^TM^ Protein Refolding Kit from AthenaES^TM^ (Athena Environmental Sciences, Inc., Baltimore, MD, USA) and containing 100 mM Tris-HCl, pH 8.9, 200 mM NaCl, 100 mM KCl, 2 mM MgCl_2_, 2 mM CaCl_2_, 1 mM GSH, 0.1 mM GSSH. The sulfhydryl reagents were added to facilitate the correct S-S bonding during the domain folding. The “refolded” protein solution was dialyzed overnight against 20 mM Tris-HCl, pH 8.9, 150 mM NaCl, and concentrated again by centrifugation in an Amicon centrifugal concentrator MWCO 10 kDa (Millipore Ireland Ltd., Tullagreen, County Cork Carrigtwohill, Ireland). After centrifugation for 5 min at 12,100× *g*, the protein solution was loaded on a Superdex 200 column for size-exclusion chromatography (see below). The peak containing the protein of interest with *M_r_* of about 20 kDa (see below) was collected, concentrated up to 1.5 mg/mL, and stored at −25 °C for the subsequent analysis.

The total yield of purified recombinant TcpA determined according to [19], was ~60 mg/L of the initial culture.

### 2.5. Mass Spectrometry Analysis of the Recombinant Protein TcpA-C

Protein bands corresponding to TcpA-C were excised from the electrophoresis gel and treated independently from each other with four proteases (Trypsin and Chymotrypsin (Sigma-Aldrich, St. Louis, MO, USA), V8 and Proteinase K (Promega Corporation, Madison, WI, USA)) at 37 °C in a Thermo Mixer thermo shaker (Eppendorf, Hamburg, Germany). 5 mM CaCl_2_ was added to stabilize proteinase K. The molar ratio of enzyme/protein was 1/50. The protein hydrolysis was stopped by adding trifluoroacetic acid. The resulting peptides were mixed and analyzed by high resolution tandem mass-spectrometry, after preliminary separation. The separation was performed by reversed-phase liquid chromatography in a gradient of acetonitrile from 4% to 60% for 120 min using an Easy nLC 1000 nanoliquid chromatograph (Thermo Fisher Scientific, Waltham, MA, USA) and a column packed under laboratory conditions with Aeris Widepore XV-C18 phase 3.6 μm particle size and 300 Å pore size. Mass spectra of the sample were obtained using an Orbi Trap Elite mass spectrometer (Thermo Fisher Scientific, Waltham, MA, USA). The peptides were ionized by electrospray at the nano-liter flow rate with 1.8 kV ion spray voltage and a capillary temperature of 200 °C. The fragmentation of ionized peptides was performed by the CID method in a high-energy cell (HCD).

The mass spectra were processed, and the peptides were identified using the PEAKS Studio (ver. 7.5) program. 111 TcpA-C peptides were identified by mass spectrometry analysis, and 106 among them appeared unique. The tolerance errors for parent mass and fragment mass were 5.0 ppm and 0.5 Da, respectively. Four variable modifications were used (o—oxidation of Met, d—dioxidation of Met, f—formylation and c—carbamidomethylation of Cys). The protein search was performed using the Uniprot database for bacterial proteins. The database contained 982,400 aminoacid sequences. The TcpA protein proved to be the major one, while others were in trace quantities.

### 2.6. Structural Characterization of the Recombinant Protein TcpA-C

All measurements (except the temperature dependence detection) were taken at room temperature.

The TcpA-C absorption spectra were recorded using a Cary100Bio spectrophotometer (Varian Medical Systems, Palo Alto, CA, USA). Protein concentration was measured using the molar extinction coefficient E_280nm_ = 3105 M^−1^cm^−1^ evaluated according to [20]. The fluorescence spectra were measured on a Cary Eclipse spectrofluorimeter (Varian Medical Systems, Palo Alto, CA, USA) with 1 × 1 × 4 cm quartz cells.

Spectra of Far-UV circular dichroism were recorded using a Chirascan Spectropolarimeter (Applied Photophysics Ltd., London, UK) at a protein concentration of 0.2 mg/mL and a quartz cell of 0.1 mm pathway length.

The TcpA-C secondary structure content was assessed using the BeStSel web server [21], CDNN CD deconvolution software 2.1 [22], K2D3 web server [23], DSSP [24], and SELCON [25] programs.

Transverse urea gradient electrophoresis was used for protein refolding/unfolding studies, as described by Creighton [26] and Goldenberg [27]. The gels were prepared using a 0.0–8.0 M urea gradient in 20 mM Tris-HCl, pH 8.9, buffer, and the samples were running for 4 h. After electrophoresis, the gels were stained with Coomassie brilliant blue G-250 and analyzed.

The size-exclusion chromatography was performed using an HPLC chromatograph ProStar (Varian Medical Systems, Palo Alto, CA, USA), a Superdex 200 10/300 GL column, and a flow rate of 0.4 mL/min. Protein detection was performed by tyrosine intrinsic fluorescence (excitation wavelength 275 nm, emission wavelength 315 nm).

The ^1^H NMR monodimensional spectrum of TcpA-C was measured using a Bruker Avance III HD 600 MHz NMR spectrometer (Bruker, Billerica, MA, USA). The working frequency was 600 MHz, with a spectrum width of 24 ppm and a 90° pulse of 11 µs. The number of accumulations was 150 at a protein concentration of 0.8 mg/mL and a temperature of 298 K. The water signal repression was performed using WATERGATE with a delay before 90° pulse of 1–2 s.

The temperature dependence of the protein partial heat capacity C_p_(T) was recorded using a SCAL-1 differential scanning microcalorimeter (Scal, Pushchino, Russia) [28] with 0.3 mL glass cells at a protein concentration of about 1 mg/mL and a heating rate of 1 K/min.

The quantity of free SH groups was defined using Ellman’s reagent [29].

## 3. Results

### 3.1. Biosynthesis of TcpA-C in E. coli Cells and Its Purification

From SDS-PAGE data on TcpA-C biosynthesis and purification (Figure 1), the following conclusions can be drawn. First, the recombinant protein with a molecular weight close to that of TcpA-C is synthesized after IPTG induction, i.e., as a product of the expression of the constructed plasmid (lane 2). Second, the bulk of TcpA-C is detected in the precipitate (cf. lanes 3 and 4) as the major protein (cf. lanes 2 and 3). Third, the protein isolated from the soluble fraction by Ni-NTA affinity chromatography seems to be sufficiently electrophoretically pure (see lanes 4 and 5). Moreover, as confirmed by mass spectrometry analysis, the bands corresponding to proteins with molecular weights close to 18 kDa include TcpA-C (see Section 2, and Appendix A).

### 3.2. Physico-Chemical Characterization of the Recombinant Protein

To develop the protocol for the large-scale recovery of recombinant TcpA-C from inclusion bodies, we used the results of transverse urea gradient electrophoresis (TUGE). Monitoring the cooperative (S-like) unfolding or refolding transitions is a strategy to characterize the rigidity of a protein tertiary structure packing (i.e., the proper protein folding) [30]. In contrast to spectroscopic and other physicochemical methods, TUGE allows monitoring the change in protein electrophoretic mobility reflecting the change in the protein hydrodynamic volume during unfolding or refolding using a small sample (~50 µg) for a short time (~4 h).

Figure 2 shows the electrophoretic pattern of renaturation of a sample extracted from cell debris after its dissolution in a buffer containing 20 mM Tris-HCl, pH 8.9, 8.0 M urea, followed by centrifugation (Figure 2a), and that of denaturation of a sample isolated from the soluble fraction by Ni-NTA chromatography (Figure 2b). It should be emphasized that the recombinant TcpA-C, the major component of inclusion bodies, undergoes a relatively abrupt refolding in 3.0–1.0 M urea. Importantly, the refolding of TcpA-C from inclusion bodies is very similar to unfolding of this protein isolated from the soluble fraction after Ni-NTA chromatography (Figure 2b).

This means that the urea-induced folded-to-unfolded transition of the recombinant protein is a spontaneous and reversible process following the two-state model:(1)$$N\overset{K}{↔}U$$

We derived the thermodynamic information from electrophoretic data by applying the simplest linear extrapolation method (LEM) to the S-shaped transition curves [31]. The electrophoresis data were digitalized (Figure 2) and fitted to the equation
(2)$$\Delta G=\Delta G_{w}-mC_{ur}$$
where
(3)ΔG=RTlnK
(4)Cur,m=ΔGwm

Δ*G* is the Gibbs energy change as a function of temperature *T* (298K in our case) and urea molar concentration *C_ur_*, while Δ*G_w_* is the Gibbs energy change in water (kJ/mol), *m* is the proportionality coefficient (in kJ/M mol), and *R =* 8.314 kJ/K mol is the gas constant. *C_ur,m_* is the denaturant concentration at the transition midpoint, which visually corresponds to the infliction point of the sigmoid curve. The average thermodynamic parameters obtained from the fitting analysis of TUGE data are Δ*G_w_ =* 12.6 ± 0.6 kJ/mol; *m =* 6.2 ± 0.2kJ/M mol, and *C_ur,m_ =* 2.0 ± 0.1 M, which correspond to those for 20 kDa globular proteins [32].

The TUGE data (Figure 2) also hint at a possibility of refolding of TcpA-C from 8.0 M urea without essential aggregation using simple dilution with the native buffer to a final urea concentration below 1.0 M. Figure 3a presents a size-exclusion chromatography profile of TcpA-C purified from inclusion bodies and refolded in the buffer 20 mM Tris, pH 8.9, 150 mM NaCl (see Section 2). As seen, the bulk of the refolded protein eluates like a globular protein with a molecular weight of about 20 kDa (Figure 3c) and an elution volume very close to that of the native protein isolated from the soluble fraction (Figure 3b), though minor aggregation is present (Figure 3a, arrow).

Figure 4 presents some physicochemical properties of the recombinant TcpA-C after its recovery from inclusion bodies (see Section 2). Figure 4a shows SDS-PAGE of inclusion bodies at different stages of purification. Lane 2 in Figure 4a represents washed inclusion bodies (see Section 2); it has one major protein band corresponding to TcpA-C by molecular weight, but the protein purity is not high enough. Further purification of TcpA-C involves Ni-NTA chromatography in 8.0 M urea (Figure 4a, lane 3) and MonoQ chromatography, followed by protein refolding and size-exclusion chromatography (Figure 4a, lane 4). The additional purification by MonoQ chromatography removes some impurities that remained after Ni-NTA chromatography at pH 8.9 within the region of 37 kDa (Figure 4a, lane 2–3). The UV-absorbance and fluorescence spectra (Figure 4b) of TcpA-C recovered from inclusion bodies (Figure 4a, lane 4) are typical for the protein chain with a high content of phenylalanine residues, only a few tyrosines, and no tryptophanes, which is in agreement with the amino acid composition of TcpA-C (Appendix A, [11,17]) and confirms the absence of highly absorbing impurities like nucleotides.

The Far-UV CD spectrum of TcpA-C recovered from inclusion bodies (Figure 4c) is typical for an α + β globular protein and is very close to that of TcpA-C purified from the soluble fraction (Appendix A). These data hint at the correct folding of TcpA-C recovered from inclusion bodies. Moreover, we compared the protein secondary structure contents in solution with that in a crystal [16] using several programs of deconvolution (BeStSel web server, CDNN CD deconvolution software 2.1, K2D3 web server, see Section 2). The result is presented in Appendix A. As seen, in the crystal and solution, α-helix and β-structure contents differ depending on the program of deconvolution, likely due to the imperfection of deconvolution of this sort of protein secondary structure. Nevertheless, we can conclude that almost half of the protein sequence is involved in the secondary structure both in solution and in the crystal.

The rigid packing of TcpA-C recovered from inclusion bodies was confirmed by differential scanning calorimetry (DSC) and ^1^H-NMR. The former is presented in Figure 5. The DSC curve fitting based on the two-state unfolding model [33] gave a very good result showing the following parameters: the transition midpoint *T_m_* = 315.9 ± 0.2 K; the heat effect Δ*H_m_* = 339 ± 10 kJ/mol; the heat capacity increment Δ*C_p,m_* = 8.5 ± 0.4 kJ/K mol at the midpoint; and Gibbs energy change at 298.15 K Δ*G*_298.15*K*_ = 14.0 ± 0.7 kJ/mol. The Gibbs energy changes revealed by two different methods (cf. Δ*G* values for isothermal unfolding by urea and those of temperature-induced unfolding) are very similar and typical of the middle-size globular protein unfolding [32].

Figure 6 represents the high field region of ^1^H-NMR spectrum of TcpA-C recovered from inclusion bodies. The presence of the high-field signals of methyl protons caused by persistent interactions between methyl groups in aliphatic residues and aromatic side chains is indicative of rigid packing of the protein tertiary structure [34].

## 4. Discussion

The main aim of the present work is to obtain the recombinant toxin-coregulated pilus A protein encoded by *tcpA^CIRS^* (TcpA) that is a promising tool for the development of new cholera vaccines [7,8]. Regretfully, the recombinant TcpA produced by expression of the appropriate plasmid in *E. coli* cells is mostly localized to intracellular aggregates called inclusion bodies [35,36] (Figure 1).

The recovery of recombinant proteins from inclusion bodies is a challenge in their large-scale production by cell biotechnology [37]. The difficulty arises from the absence of prediction of the protein ability to refold after denaturant-induced unfolding [38,39]. Here, we used a simple and efficient approach to determining this ability after the solubilization of inclusion bodies in urea-containing buffers. This approach, the so-called TUGE, was previously developed by T. Creighton and D. Goldenberg [26,27] for visualization of unfolding/refolding transitions of globular proteins. TUGE allows tracking changes in mobility of the major protein component of cell lysate precipitate with decreasing urea concentration and requires neither high purity nor large amount of the substance (Figure 2a). Additionally, this method allows defining the urea concentrations for the native and unfolded states of proteins and for possible aggregation of intermediate conformations. Moreover, it provides some thermodynamic parameters of the protein stability and the information on rigidity of the refolded protein tertiary structure. The presence of even a small amount of the recombinant protein in the soluble fraction of *E. coli* lysate (Figure 1) shows that this protein can be folded in cytoplasmic conditions (pH, ionic strength, temperature), and allows comparison of some structural parameters of an in vivo folded protein with those of the protein in vitro recovered from inclusion bodies (Figure 2, Figure 3 and Appendix A). We demonstrate that the renaturation curve of TcpA whose aggregates are dissolved in urea (Figure 2a) and the denaturation curve of this protein isolated from the soluble fraction (Figure 2b) are very similar, and their processing gives almost identical thermodynamic parameters. Furthermore, the S-like change in protein electrophoretic mobility is typical for the urea gradient-induced folding/unfolding transition of a rigid globular protein [26]. The decrease in urea concentration resulting from 20-fold dilution with the native buffer, followed by size-exclusion chromatography, leads to refolding of TcpA into a globular conformation similar in hydrodynamic dimension to that of the protein isolated from the soluble fraction of the cell lysate (Figure 3). Moreover, Far-UV CD spectra of these proteins are very similar to each other (Appendix A), thus indicating the similarity of their secondary structure contents. Although, in our case, the TcpA-C production was hampered by formation of inclusion bodies, the yield of pure protein was evaluated as 60 mg/L of the initial culture. This value is intermediate between previously reported yields of TcpA from other strains, e.g., 8 mg/L [7] and 2.1 mg/mL [8]. The large-scale recovery of TcpA from inclusion bodies allows its characterization by several physico-chemical methods requiring a significant amount of protein. Thus, the circular dichroism spectra (Figure 4c) indicate a high extent of recovery of the protein secondary structure. The presence of a rigid tertiary structure in TcpA recovered from inclusion bodies is clearly demonstrated by the presence of a pronounced peak on the DSC heat absorption curve (Figure 5). Moreover, Gibbs free energy determined by isothermic unfolding/refolding of TcpA (Figure 2), coincides with the value determined by DSC (Figure 5). Note that the rigidity of the tertiary structure of recombinant TcpA-C recovered from inclusion bodies according to the proposed protocol is confirmed by the presence of high field resonances of methyl protons in the ^1^H-NMR spectrum (Figure 6). Unfortunately, the physico-chemical parameters of the recombinant TcpA-Cs from other strains are underexplored, which prevents comparison of different TcpA-C preparations [7,8,10]. The exception is crystallographic studies of the TcpA^ET^-C globular domain (Appendix A) [16], which give a detailed information about the protein spatial structure. One interesting piece of data is the presence of the single intramolecular disulfide bond between Cys145 and Cys211 (Appendix A) [16]. This feature hints at the possibility of forming S-S bonds in *E. coli* cytoplasm and highlights an important point indicating the correct protein folding. Using Ellman’s reagent [29], we ascertain that our preparation of TcpA-C recovered from inclusion bodies contains an S-S bond. Moreover, the mobility of TcpA-C in SDS-PAGE is visibly reduced after protein treatment with dithiothreitol (DTT) (Appendix A), likely due to an increase in the protein hydrodynamic volume caused by disruption of the disulfide bond.

## Figures and Tables

**Figure 1 life-12-01802-f001:**
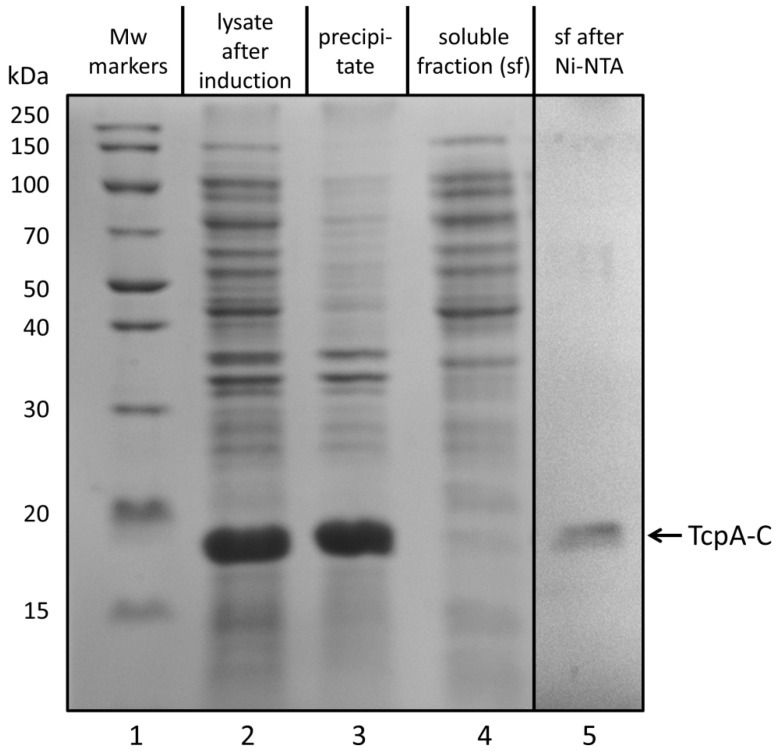
SDS-PAGE of TcpA-C purification: molecular weight markers (lane 1); cell lysates of *E. coli* after IPTG induction (lane 2); proteins in the precipitate (lane 3) and in the soluble fraction (lane 4); the soluble fraction after Ni-NTA affinity chromatography (lane 5). Lane 5 originated from another gel was corrected according to the lane of molecular weight markers.

**Figure 2 life-12-01802-f002:**
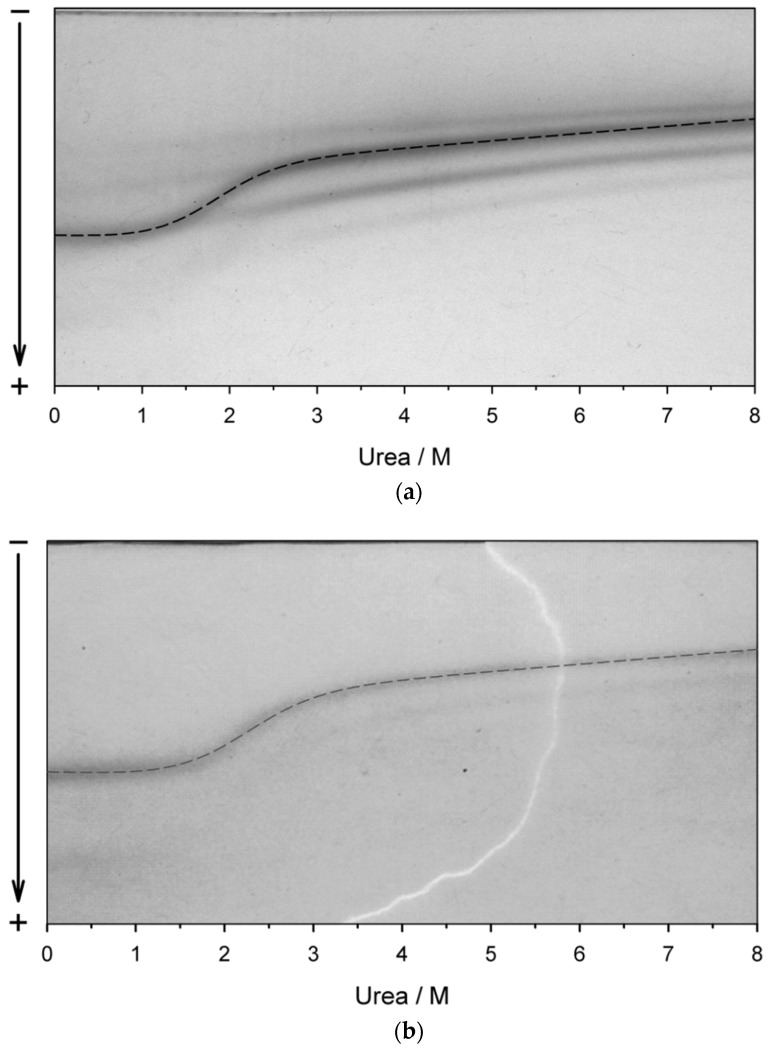
Transverse urea gradient electrophoresis of (**a**) protein components of the cell precipitate dissolved in 20 mM Tris-HCl, pH 8.9, 8.0 M urea, followed by centrifugation (refolding); (**b**) unfolding of TcpA-C isolated from the soluble fraction after Ni-NTA and Superdex 200 chromatography. The dashed lines show the points used in the evaluation of thermodynamic parameters.

**Figure 3 life-12-01802-f003:**
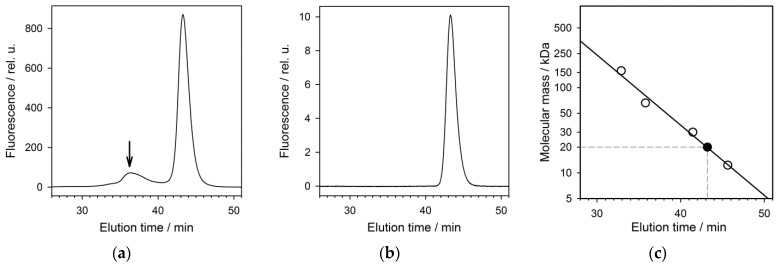
Size-exclusion chromatography on Superdex 200 (20 mM Tris-HCl, pH 8.9, 150 mM NaCl): (**a**) TcpA-C purified from inclusion bodies and refolded; the concentration of the loaded protein was 1 mg/mL; an arrow indicates the peak corresponding to aggregates; (**b**) TcpA-C from the soluble fraction after Ni-NTA chromatography; the concentration of the loaded protein was 0.05 mg/mL; (**c**) the dependence of the elution time on the protein molecular weight (the chromatographic column calibration); proteins used for the calibration are aldolase (158 kDa), bovine serum albumin (66 kDa), bovine carbonic anhydrase B (29 kDa), and cytochrome *c* (12.3 kDa), the positions of which are shown by open circles, while the position of monomeric TcpA-C is indicated by a closed circle.

**Figure 4 life-12-01802-f004:**
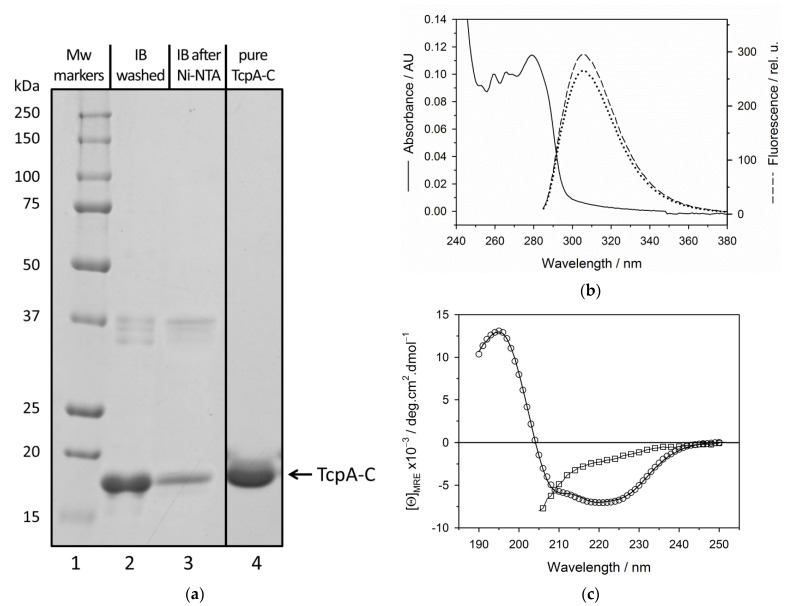
Physico-chemical properties of the recombinant TcpA-C. (**a**) SDS-PAGE of TcpA-C from inclusion bodies (IB): molecular weight markers (lane 1); washed IB (lane 2); TcpA-C after purification from washed IB by Ni-NTA chromatography in 8.0 M urea (lane 3), followed by MonoQ chromatography and refolding (lane 4); the position of lane 4 originated from another gel was corrected according to the lane of molecular weight markers; (**b**) Far-UV absorption (solid line) and intrinsic fluorescence (dashed line) spectra of TcpA-C purified and refolded from inclusion bodies (20 mM Tris-HCl, pH 8.9, 150 mM NaCl), and intrinsic fluorescence (dotted line) spectrum of TcpA-C unfolded in 8.0 M urea; (**c**) Far-UV CD spectra of TcpA-C purified and refolded from inclusion bodies (20 mM Tris-HCl, pH 8.9, 150 mM NaCl) (circles), and of TcpA-C unfolded in 8.0 M urea (squares).

**Figure 5 life-12-01802-f005:**
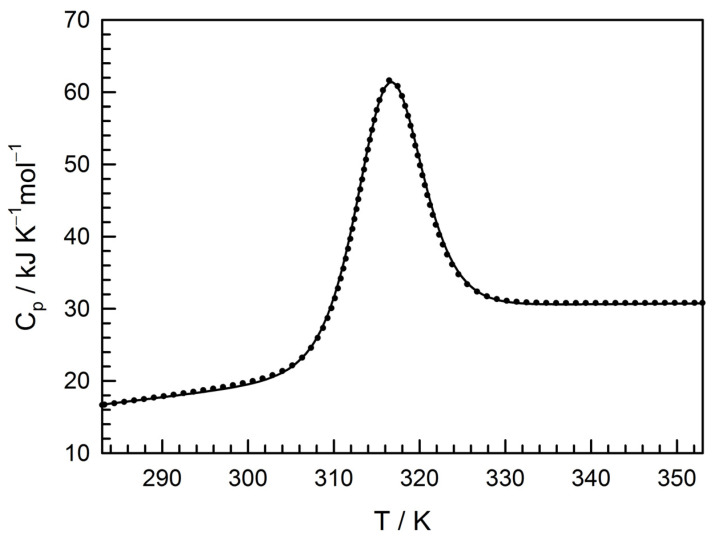
Temperature dependence of the molar heat capacity for TcpA-C purified and refolded from inclusion bodies at pH 8.9, 20 mM borate buffer. The solid line denotes the experiment; the dots show the best fitting to the two-state model.

**Figure 6 life-12-01802-f006:**
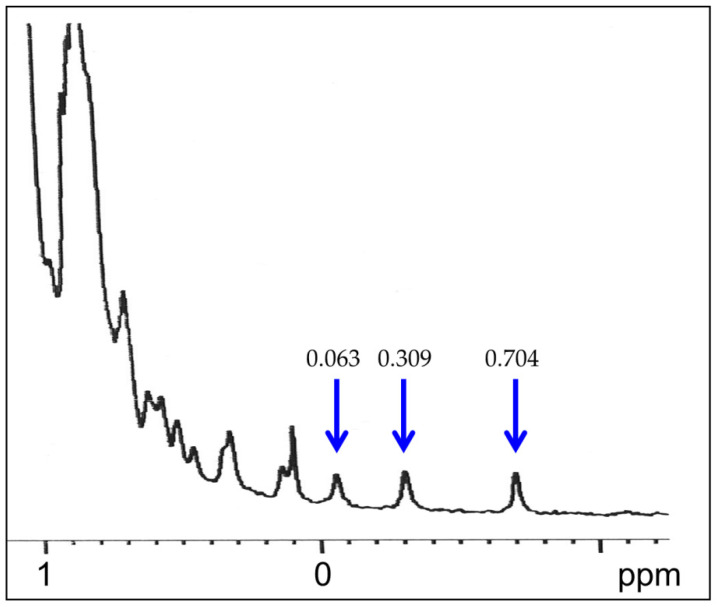
The high field region of the ^1^H-NMR spectrum of TcpA-C recovered from inclusion bodies. The arrows indicate the high field signals of the methyl group protons.

## Data Availability

Not applicable.

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
