# Peer review of "Recombinant Globular Domain of TcpA Pilin from Vibrio cholerae El Tor: Recovery from Inclusion Bodies and Structural Characterization"

_life, 2022, doi:10.3390/life12111802_

Round 1
Reviewer 1 Report
This manuscript by Marchenkov et al. describes the folding of TcpA-C protein from inclusion bodies dissolved in urea using TUGE and biophysical parameters. I have the following comments and suggestions for the manuscript.
1. The unfolding/refolding of TcpA-C was visualized using TUGE and some parameters are represented. In addition to these, the recovery of tertiary structure could be verified using biochemical assays to figure out function of TcpA-C.
2. p.2 line 65 : Does TcpA-C mean the globular C-terminal domain of TcpA
? The explain about TcpA-C is needed when it first appear.
3. p.2 line 66 : The citation [9] regarding of the crystal structure determined at a resolution of 1.3 Å is inappropriate. The resolution of structure in [9] is 1.5 Å. Please check it.
4. p.2 line 84 : modify the celsius scale unit of 30 oC
5. p.4 line 158: SDS PAGE —> SDS-PAGE
6. p.5 line 174, 177, 178 : The unity of decimal places of urea concentration is needed. 8.0 M, 3-1.0 M, ~2 M
Besides these, please check the concentration of urea in other pages.
7. p.6 Figure 3. (a)/(b) Y axis : usually, the result of size-exclusion chromatography of protein was represented with the absorbance in AU unit according to time. please check it.
8. p.8 Table 1. To compare the secondary contents of twoTcpAs using CD, the measurements should be conducted in same condition. however, this result seems to be conducted in separatley. Additionally, the α /β contents are also different.
9. p.8 line 264 : Escherichia coli —> E. coli
10. p.9 line 272 : transverse urea gradient electrophoresis (TUGE) —> TUGE
11. p.9 line 305 : 1H —> 1H
12. p.5 Figure 2b : change the SDS-PAGE figure without crack.
Reviewer 2 Report
The manuscript of Marchenkov et al. describes the recombinant production of TcpA-C from V. cholerae from inclusion bodies made in E. coli. The authors are able to use transverse urea gradient electrophoresis to inform a solubilization and refolding protocol and achieve successful refolding and 2-step purification of the target protein with only minimal amounts of aggregates present in the final preparation. Some physicochemical properties are further compared to a soluble preparation of TcpA which is produced by E. coli at around 10% efficiency. While I think the manuscript deserves to be published and can be of value for other researchers aiming to produce larger amounts of TcpA, there are a few points that need to be addressed (please see my comments below). In particular, the value of several experiments could have been augmented by including ‘negative controls’ where the refolded protein is compared to an intentionally denatured sample to clearly show differences rather than just remarking on the resemblance to what would be expected from a folded protein. I also think the manuscript would benefit from some English language editing. While orthography is perfectly fine, sentence structures may be improved in several paragraphs.
Line 55: cleaved off rather than cut off
Lines 85-86: Could the authors please indicate the IPTG concentration used for induction and the exact expression conditions in terms of time and temperature? These variables seem essential in the context of the paper as they may influence the aggregation propensity (and thus the tendency to form inclusion bodies) of a heterologous protein. These parameters may be given in reference 10, but since this publication is in Russian, evaluation is not straightforward.
Line 90: lysed instead of destroyed might be a better term here
Lines 96-101: Again, could the authors please give a brief description of the purification protocol in the manuscript? Reference 10 was only available to me in Russian.
Lines 113-120: This part of the method section reads like the result section. Please limit descriptions in this section to the methodology only.
Lines 197: Please change twist point to infliction point
Lines 224-225: It would be helpful if the authors could briefly explain the characteristics of the UV-Vis and fluorescence spectra that led them to their conclusion. Also, I’d generally question the value of presenting a single fluorescence spectrum in this case. A comparison of the refolded protein in a buffer with and without high urea concentration would clearly demonstrate the success of refolding with this method. This is also true for the CD-spectroscopy experiment in which the spectrum of the refolded protein could have easily been compared to one of a re-denatured one.
Line 236: Please specify the software used for calculating the secondary structure content from the CD spectrum.
Figure 5: It would be interesting to see a comparison of the protein recovered from inclusion bodies with the one recovered from the soluble fraction. Possibly, the amount of protein recovered from the soluble fraction is insufficient for a DSC experiment. However, the authors may consider performing a comparative differential scanning fluorometry (DSF) which requires only microgram amounts of sample to demonstrate similar thermal stability of the two preparations.
Lines 303-306: It may be tempting to discuss 1H-NMR results in a manuscript, but in order to use them to confirm experimental results, I’d ask the authors to at least show the spectrum in a supplemental figure or to remove this statement. There is simply no way for the reader (and the reviewer) to examine this claim without seeing the spectrum. Further, I believe that adding 1D NMR spectra may considerably increase the value of this study.
Discussion section: It would be interesting to read a (short) paragraph about previous studies that produced TcpA to judge better the value this study adds to the field. In particular, it may be interesting to know whether the protein has been successfully produced in E. coli strains capable of making S-bonds in the cytoplasm (e.g Shuffle).
Reviewer 3 Report
The manuscript by Marchenkov et al., describe the denaturing recovery from inclusion bodies and subsequent purification of the C-terminal domain of the V. cholerae TcpA pilin protein. The motivation for the work is to develop a large-scale purification method to recover TcpA-C from inclusion bodies for use in downstream biotechnology applications such as antibody development and immunodiagnostics. This manuscript describes the initial purification protocol and minimal biophysical characterization to probe the secondary and tertiary structure of the recovered protein. As the manuscript stands now, there are major deficiencies that should be addressed before the manuscript is suitable for publications.
Major Comments
1. The rationale for the work is unclear, if a purification method exists that is capable of producing large quantities of high-quality protein (for crystallography and structure determination), why w is it necessary to develop an alternate protocol for the same protein?
2. How does the protocol presented in this work compare to the protocol used in the crystallography manuscripts in terms of yield, protein stability, structure, correct fold etc.
3. The authors report a partial sequence of their construct – how does this compare to the published structure? A sequence comparison between the two constructs and full-length wild-type TcpA (El Tor) would be instructive. The sequence differences between the authors constructs with those that have been previously reported should be highlighted.
4. Figure 1 – lane 3, it is questionable that these are in fact inclusion bodies in the purest use of the term, this looks more like the insoluble fraction of the cells. This is similarly supported by the authors who state that they remove the contaminating proteins from the inclusion bodies with an Ni-NTA column in 8 M urea. These observations indicate that TcpA-C is not expressed in inclusions.
5. The contrast for lane 5 is different from lanes 1-4. It needs to be indicated that these are not from the same gel.
6. Please label panels in the upper left corner, including the panel labels below is distracting
7. In the analysis of the TUGE experiment, how are the authors sure that the TcpA-C recovered from the soluble fraction is correctly folded?
8. The explanation for the differences on secondary structure content between the deconvoluted CD spectra and calculated from the crystal structure is insufficient. Frankly, if the authors purified protein spontaneously interconverts almost 50% of its secondary structure, the authors have not purified the protein they think they have purified. This is not a result that should be dismissed.
9. The solution conditions (temperature, pH, buffer, salt) for the biophysical characterizations (CD, DSC, SEC) are not reported in the figure captions or in the methods section. How was the protein stored after purification and before analysis? Was it frozen, refrigerated? What was the concentration of the stock solution?
10. The authors suggest that methyl resonances in the 1H NMR spectra are evidence of correct folding but without resonance assignments the authors really don’t know if these arise from folded residues or not. Further, from the CD data, the authors clearly have folded protein but what is at issue is if the protein is correctly folded.
11. The authors position their method as an approach for “large-scale production” for biotechnology applications such as vaccine production. What is the average yield from their approach and how does this compare with methods that have already been reported?
12. The discussion reads as a recap of the results section and needs to be reorganized and rewritten. What is important for the authors to make the results of their manuscript more convincing is a direct comparison between protein purified with the method used to produce protein for the crystallographic studies of TcpA-C. The biophysical measurements should be repeated on this standard and directly compared with their refolded protein. The authors should also perform intact mass analysis to confirm that they are in fact purifying the correct construct (and that their construct is expressed correctly and not being truncated). Also, since the authors are suggesting that their refolded protein is correctly folded relative to the crystal structure, an 1H,15N HSQC (or similar correlative experiment) would enable the authors to definitively declare if their refolded protein adopts the same structure as the reported structure.
Minor Comments
Line 49 – “it has 49 been proved” to “it has been proven”
Line 55 – “to the signal that” to “to the signal peptide that”
Line 109 – “to test an ability for a major population” to “to test if a major population of Tcp-A can be refolded”
Line 123 – “till” to ‘to”
Line 158 – “”allow the” to “allow for the”
Figure 1 caption – consistent formatting for the lane descriptions. It would be useful to label the lane contents directly on the figure.
Line 236 – which program?
Line 241 – “contents” to “content”
Through proof reading of the manuscript to address grammatical and typographical errors
Reviewer 4 Report
This manuscript describes the purification of a portion of the pilin protein from the Vibrio Cholarae strain El Tor. The protein predominantly expresses as inclusion bodies, but the authors describe a high-efficiency refolding method to produce soluble protein. They also conduct some biophysical characterization on the refolded protein for validation. Overall, this is a decent paper and a good body of work but there are a few weaknesses detailed below:
Major issues:
1. There is not conclusive evidence that the expressed protein is the correct sequence. The authors present SDS-PAGE gels that show a band for a protein that appears to be overexpressed but there is no untransformed control to verify that this is the case. There is also no Mass Spec or amino acid composition or some other measure that could verify the correct protein has been expressed.
2. The CD data calls into question whether this protein is folded correctly. The authors did a deconvolution of their spectra and compared those results to the known crystal structure percentages of secondary structure. However, these values did not agree very closely. It seems like a better comparison would have been to the soluble protein from the expression since this would show that the protein refolded to the correct conformation. These spectra should overlay if refolding occurred correctly, and it eliminates the need for justifying the observed discrepancies.
3. The authors discuss on lines 245-247 that “Other protein components . . . appear to aggregate at about 2.0 M or lower urea concentration.” It is not clear how the authors reached this conclusion.
4. There are several places where the manuscript was hard to interpret or need revision. These include:
a. A single sentence should not be alone (lines 42-43). This sentence should be included in the first paragraph or the second paragraph. Similarly, the sentence on lines 237-238 should not be separate.
b. The results section begins with a description of an SDS-PAGE gel with no introduction to what the gel is about or why it is being included. A short introduction to the goals of the project would help the reader understand why the gel is being included. This occurs repeatedly throughout the Results part of the paper.
c. The authors discuss results from TUGE without any discussion of the technique or why it is being used. This is not a common technique so some of the language from the Discussion would be appropriate here or in the introduction to help the reader who is unfamiliar with the technique understand the results.
d. There is new data introduced in the Discussion in the form of the NMR experiments. This should be in the results section and not here. Also, the figures in the Supplemental Data should be in the order that they appear in the manuscript so Figure S4 should be before Figure S3.
Minor Issues:
There are places where the language could be improved to help the reader. For example, “a” makes more sense than “the” on lines 26-27. The paper would be much better with a very careful proofread from a native English speaker.
Round 2
Reviewer 3 Report
The authors have addressed the majority of my technical concerns. The rationale behind the study remains weak. According to the authors, the only practical difference between the ET and CIRS constructs is the N42S mutation (authors construct numbering) and therefore they could not express the ET construct because they do not have access to the Origami strain. There are other similar strains of E. coli (e.g. Shuffle) and the authors clearly have access to modern biochemical supplies and instrumentation (FPLC, columns, spin concentrators) etc. used in this work. It is hard to believe that they don't have access to these cells. Additionally, they report a denaturing purification and in vitro refolding protocol for recovery from inclusion bodies. Are the Origami cells really necessary? My assessment remains, this work is not novel or significant enough for publication and would be better suited to be included as part of a larger work describing at the very least the difference between the ET and CIRS proteins.
Reviewer 4 Report
The authors have addressed my concerns adequately.
Round 3
Reviewer 3 Report
n/a